| 1  | Haze pollution in winter and summer in Zibo, a heavily industrialized city                                                                                                         |
|----|------------------------------------------------------------------------------------------------------------------------------------------------------------------------------------|
| 2  | neighboring the Jin-Jin-Ji area of China: source, formation, and implications                                                                                                      |
| 3  | Hui Li <sup>a</sup> , Fengkui Duan <sup>a*</sup> , Yongliang Ma <sup>a</sup> , Kebin He <sup>a*</sup> , Lidan Zhu <sup>a</sup> , Tao Ma <sup>a</sup> , Siqi Ye <sup>a</sup> , Shuo |
| 4  | Yang <sup>a</sup> , Tao Huang <sup>b</sup> , Takashi Kimoto <sup>b</sup>                                                                                                           |
| 5  | <sup>a</sup> State Key Joint Laboratory of Environment Simulation and Pollution Control, School of                                                                                 |
| 6  | Environment, State Environmental Protection Key Laboratory of Sources and Control of Air                                                                                           |
| 7  | Pollution Complex, Beijing Key Laboratory of Indoor Air Quality Evaluation and Control,                                                                                            |
| 8  | Tsinghua University, Beijing 100084, China                                                                                                                                         |
| 9  | <sup>b</sup> Kimoto Electric Co. Ltd, Funahashi-Cho, Tennouji-Ku Osaka 543-0024, Japan                                                                                             |
| 10 | *Corresponding authors.                                                                                                                                                            |
| 11 | E-mail: duanfk@tsinghua.edu.cn, hekb@tsinghua.edu.cn.                                                                                                                              |
| 12 | Phone: +86-10-62797900                                                                                                                                                             |

13 Fax: +86-10-62797900

### 14 Abstract

Continuous field observations of haze pollution were conducted in winter and summer during 15 2015 in Zibo, a highly industrialized city in the North China Plain that is adjacent to the Jing-Jin-16 Ji area. PM<sub>2.5</sub> concentration averaged 146.7  $\pm$  85.8 and 82.2  $\pm$  44.3 µg m<sup>-3</sup> in winter and summer, 17 respectively. The chemical component contributions to PM<sub>2.5</sub> showed obvious seasonal variation. 18 Organics were high in winter, but secondary inorganic aerosols (SIA) were high in summer. From 19 20 non-haze to haze days, the concentration of SIA increased, implying an important role of secondary processes in the evolution process of the pollution. The diurnal behavior of several 21 22 pollutants during haze days appeared to fluctuate more, but during non-haze days, it was much more stable, suggesting that complex mechanisms are involved. Specifically, gaseous precursors, 23 mixed layer height (MLH), photochemical activity, and relative humidity (RH) also played 24 important roles in the diurnal variation of the pollutants. Normally, larger gaseous precursor 25 concentrations, photochemical activity, and RH, and lower MLH favored high concentrations. In 26 27 winter, the formation of sulfate was mainly influenced by RH, indicating the importance of heterogeneous reactions in its formation. In contrast, in summer, photochemistry and SO<sub>2</sub> 28 29 concentration had the largest impact on the sulfate level. We found that Zibo was an ammoniarich city, especially in winter, meaning that the formation of nitrate was through homogeneous 30 reactions between HNO<sub>3</sub> and NH<sub>3</sub> in the gas phase, followed by partitioning into the particle 31 phase. The RH, NO<sub>2</sub>, and "Excess NH<sub>4</sub><sup>+</sup>" were the main influencing factors for nitrate in winter, 32 33 whereas "Excess NH4<sup>+</sup>", RH, and temperature were the key factors in summer. The secondary 34 organic carbon (SOC) level depended on the MLH and photochemistry. In winter, the effect of the MLH was stronger than that of photochemistry, but a reversed situation occurred in summer 35 36 because of the intensive photochemistry. Our work suggested that the inter-transport between 37 Zibo, one of the most polluted cities in north China, and its adjacent areas should be taken into

- 38 account when formulating air pollution control policy.
- 39 Keywords: haze days; secondary inorganic aerosol; mixed layer height; regional air transport

## 40 1. Introduction

Airborne particles have adverse impacts on human health (Fang et al., 2016), lead to reductions 41 of visibility (Zhao et al., 2011), and play a role in global climate (Woo et al., 2008), and this is 42 43 especially so for PM<sub>2.5</sub>, particles with an aerodynamic diameter smaller than 2.5 µm (Janssen et al., 2011). Particle pollution is caused by a wide variety of anthropogenic emissions, such as those 44 45 from coal combustion, vehicles, and chemical (and other) industries, and it has become a major 46 environmental issue in recent years. To address this problem better, more complete knowledge about the particle sources, chemical characteristics, and formation processes is necessary. To 47 present, many scholars have carried out research on PM2.5 pollution in China from various aspects. 48 For example, Hu et al. (2012) investigated the features and sources of carbonaceous matter at 49 Back Garden, a rural site 50 km northwest of Guangzhou, using a semi-continuous thermal-50 optical carbon analyzer. Andersson et al. (2015) characterized the combustion sources during the 51 52 January 2013 haze events over eastern China using dual carbon isotope constrained source apportionment methods for the North China Plain (NCP), Yangtze River Delta (YRD) and Pearl 53 River Delta (PRD). He et al. (2014) proposed that mineral dust and  $NO_x$  contribute to the 54 55 transformation of SO<sub>2</sub> to sulfate during polluted days based on smog chamber results. Han et al. (2014) adopted the Regional Atmospheric Modeling System-Community Multiscale Air Quality 56 (RAMS-CMAQ) modeling system ), coupled with an aerosol optical property scheme, to 57 58 simulate the meteorological conditions, main particle composition, and visibility over the NCP 59 in 2011. However, despite the large number of studies that have been done in China, particle pollution control in China and many other countries remains a huge challenge because of the 60 complexity in sources (Sun et al., 2014), compositions (Zhang et al., 2012), processes (Tao et al., 61 62 2014) and influencing factors (Kadiyala and Kumar, 2012).

63 The NCP is one of the most developed and most polluted regions in the world (van Donkelaar et

al., 2010), and much research has already been done to investigate the air pollution there. Sun et 64 al. (2016) characterized the evolution of haze formation on the NCP and investigated the 65 66 associated stagnant meteorological conditions, including temperature inversions, low wind speed, 67 and high relative humidity. Xu et al. (2011) carried out an observation on gaseous pollutants and meteorological parameters in Wuqing, located between Beijing and Tianjin in the NCP, from July 68 69 9, 2009, to January 21, 2010. Furthermore, according to modeling results, the summer high pressure systems make the East China Plain a "basin" of ozone pollution (Zhao et al., 2009). The 70 71 Paris Agreement, which was signed on April 4, 2016, has drawn more attention to the problem of 72 climate change worldwide. China faces a serious challenge of air pollution control, especially in the NCP, thus a wide range of research on haze pollution in the NCP area is needed. 73

In this work, we conducted a field investigation from January 15–25 and from July 14–31, 2015, 74 in Zibo, a heavy industry city in Shandong Province near the Jing-Jin-Ji area, the most polluted 75 area in China. A typical winter and summer month were studied together to obtain an improved 76 77 understanding of the characteristics of the PM<sub>2.5</sub> pollution. First, time series of pollutants, meteorological parameters, and chemical compositions are presented to provide a primary 78 79 summary of the pollution condition in the two seasons. Then, we examine the diurnal variation of various parameters, including particles, species, and meteorological factors to understand the 80 possible links and causes of the diurnal patterns. Furthermore, an analysis of the main particle 81 82 constituents is undertaken to provide a deeper knowledge of the formation mechanisms and main 83 influencing factors, which should facilitate the implementation of appropriate actions to optimize the air quality effectively. Finally, we analyze the causes of the air pollution from a regional 84 perspective. In summary, this paper aims to characterize (1) the chemical composition of PM<sub>2.5</sub> 85 86 in Zibo in both winter and summer; (2) the diurnal behavior of different pollutants; (3) the 87 formation processes of sulfate, nitrate, and organic matter (OM); and (4) the regional contribution

to the particle pollution in Zibo.

#### 89

90

91 Figure 1. Location of Zibo (green area) within Shandong Province (red area), China.

## 92 2. Methodology

93 A comprehensive field observation of atmospheric PM<sub>2.5</sub> was carried out in Zibo during typical 94 winter and summer periods, i.e., from 15 to 25 January 2015 and from 14 to 31 July 2015. The instruments were set up on the roof of a building on the campus of the Shandong University of 95 Technology in Zibo, Shandong province. The inlets of the PM<sub>2.5</sub> devices were at about 20 m 96 97 above ground level. The continuous monitoring of the hourly concentration of PM2.5 of its main chemical species and of the meteorological parameters formed the foundation for this study. 98 99 Specifically, a dichotomous monitor (PM-712; Kimoto Electric, Ltd., Japan) was employed to 100 measure the concentration of PM<sub>2.5</sub> at a flow rate of 16.7 L min<sup>-1</sup>; a more detailed description of 101 this instrument has been given in previous work (Duan et al., 2016). Water-soluble inorganic ions (SO<sub>4</sub><sup>2-</sup>, NO<sub>3</sub><sup>-</sup>), were monitored simultaneously by deploying a dichotomous aerosol chemical 102