# Peer review of "Fax: +86-10-62797900"

_Atmospheric Chemistry and Physics, 2018_

## Referee Comment (RC1) · Anonymous Referee #1 · 4 Mar 2018

The paper reported the field measurement results at an urban site in industrialized cities in North China Plain, where particle pollution is a severe problem. The authors identified several episodic events in both winter and summer, reported the chemical species and sources of PM2.5 particles, and proposed the formation mechanisms of the secondary species, especially the secondary inorganic species. Overall, the manuscript is well written. Detailed analyses of the formation of haze events, especially during different seasons over the year, are helpful for pollution control. There are a few points need to be addressed before publishing.

Major comments:

[Figure]

1, Line 111: what do the authors mean that "the final hourly ammonium was corrected with the offline data"? Please provide more information on the data correction; and is it reliable to do so? What are the uncertainties?

2, Line 154: Please provide more details on the definition of haze and non-haze days. How to categorize the haze and non-haze days?

3, Line 245-246: I'm not sure if CO can be used as an indicator since the author also mentioned in the introduction that CO is largely emitted from coal combustion. And in winter, the emission is probably larger due to heating activities. Dilution or concentration resulted from the change of MHL is not the only reason for the variation of CO. For example, as shown in Figure 4, CO increased largely from 00:00 to 8:00 am but the pressure was roughly stable. MLH does not seem to be the reason. Are there any sources of CO during the night?

4, Line 251: RH seems to peak at 9:00am instead of 11:00am as shown in Figure 4a. Please elaborate more on why photochemical and heterogeneous reactions contributed to the formation of secondary species? How the conclusion is reached?

5, (1) Lin 279- 280: As the authors state, EC is a primary pollutant; hence, it is closely related to primary emission. Primary emission can be very different during a day, i.e. increased EC emission during traffic rush hours, which will certainly result in variations. Similar to CO, the authors cannot assume concentration of EC is more sensitive to MLH. (2) Line 310: Ditto. Please provide more information on whether using EC as an indicator for dilution effect is appropriate.

6, Line 337-338: Precipitation would also decrease the concentration of other species such as EC. Is precipitation data available to support that the decrease of sulfate is due to the scavenging effect?

7, Line 372-374: "Hydrolysis of N2O5 (the production of O3 and NO2) occurs mainly at night, thus the lower concentration of O3 may be a result of its contribution to N2O5,

that is, HNO3 at night (Pathak et al., 2011b). " The conclusion is a way too speculative. The reaction has to be very significant to be the sink for O3. If so, this would conflict with the conclusion by the authors later that particles phase nitrate is formed through the gas-particle partitioning of the gas-phase formed ammonium nitrate.

8, line 548-549: It is natural to expect that air masses from coastal area is cleaner than from other places. The authors attribute the high OM concentration in S1 to biomass burning. But from the fire spots data, it seems more fire spots were shown along cluster 4, yet organic concentration in S4 is lower than that in S1. Is there any other evidence, such as metal ions or organic tracers from biomass burning to support that cluster 1 is affected by biomass activities? Coincidently, another coastal cluster, cluster 5, also results in high OM concentration compared with cluster 4 although cluster 4 has a higher wind speed.

Minor comments:

Line 17: should read "Average concentrations of PM2.5 were 146 and 82 in winter and summer respectively." Line 25: "variation"->"variations" Line 110:"from"->"by" Line 189: delete "content" Line 201: "ratio"->"fraction" Line 315: "increasing" should read "increase" Line 321: EC and sulfate-> EC or sulfate Line 372: production->product

---

## Referee Comment (RC2) · Anonymous Referee #2 · 5 Mar 2018

General comments:

In this study, the authors presented the field measurements during two relatively short periods: Jan 15-25, 2015 and July 14-31, 2015 in a heavily industrialized city in China. The authors measured PM2.5, SO4, NO3, OC, EC, as well as gases NOx, SO2, CO and O3. The authors also measured meteorological variables of wind speed, wind direction, pressure, temperature and relatively humidity. From these about 4 weeks' measurements, the authors tried to make conclusion of the source, formation of two smog events in winter, and one smog event in summer. The authors concluded that high concentration of precursor, photochemical activities, relative humidity and low mixed

layer height favor the formation of smog. The authors also claimed that photochemistry and SO2 concentration had the biggest impact on sulfate; Relative humidity, NO2, and "Excess NH4+" were the main influence factor for the formation of nitrate in winter, and some other conclusions.

Operational and routine measurements of PM2.5, SO4, NO3, NH4, SO2, HNO3, and O3 have been carried out around the world for many decades. For example, these measurements have been carried out routinely for over a hundred sites in US and Canada for over three decades. The mechanism of formation of SO4, NO3, and NH4 has been well understood, and has been implemented successfully in various air quality models. There is no need to explain these well-known knowledges again and again through another short-period campaign. I tried to find any "new" results or new methods from the paper, but I didn't find. Reading through the abstract, what I found is everything I have known through textbook. Even the well-known knowledge was not explained well by the authors using the short-period observations. In general, I didn't see any valuable new information was presented in the paper, some section was very poorly written, eg. Section 3.6 Regional contributions. I firmly suggest the paper be rejected.

Specific comments:

1. The role of meteorology in formation of smog.

The meteorological condition, coupled with emission, is the major factors in formation of smog events. In this paper, the authors provided pressure, relative humidity, wind speed and temperature. From the time series of these variables, I can't see the meteorological conditions during these four weeks. To analyze the meteorological conditions, the authors need to analyze the 500 hPa and surface weather maps; need to provide the information of precipitation. The authors claimed that mixed layer height plays a role, but I don't know how the authors have the information of mixed layer height. Did the authors analyze the temperature profiles? Otherwise, how can you claim mixed

layer height play a role. Even the authors did do a good job of analyzing the vertical temperature profile to show that the smog events are related to low mixed layer height, this is new knowledge. It is well known and obvious.

The authors provided the time series of pressure, how it is linked to air quality? How wind speed and temperature are linked to the observed pollution?

In general, I saw too many pre-assumption, instead of rigorous analysis in the paper.

2. Chemical composites of haze

Chemical composites of haze in Eastern China are mainly due to SO4, NO3, NH4 (these are due to the emission of SO2 and NOx) as well as organic matters. These are well known and have been shown in many previous publications.

3. Diurnal variation

Page 233-238 The authors linked the diurnal variation of air pollutants to air pressure, as a scientist of deep knowledge in meteorology, I don't see the justification. Page 242-246 1. It is well known stable weather condition (low wind speed, under a high pressure system as indicated by weather map, not just pressure, inverse of low-level temperature profile) favors the formation of pollution event. This kind of common knowledge has been applied operationally in air quality forecast. In the analysis, the authors used too many "maybe" instead of rigorous reasoning.

It is common that due to solar radiation at the day time, the surface is heated, and the temperature profile changes. So usually during day time, the boundary height is higher than during night. This is especially true during summer time. But this is a general situation of meteorology. For a specific synoptic event, we can also have lower boundary level height during day time than during night.

Essentially, the authors used known knowledge to explain the diurnal variation, but I didn't see any new thought or idea by the authors. Also, in explanation, the authors used too much "maybe".

4. Sulfate

Line 323-325: Here the authors looked at the correlations between EC and SO4. First, these two species are not too much linked. EC is from primary emission, and SO4 is mainly from oxidation of SO2. I didn't see a good correlation between them, even from the authors' figure 5. The reasoning in line 323-325 was totally not justified, and can be wrong. SO4 is oxidation from SO2. High value of SO4 is not necessarily associated with high value of SO2. Also we need to realize that the life time of SO4 and SO2 are very different.

Relative humidity can be associated with many different meteorological conditions. For example, high relative humidity can be more associated with cloudy days, which mean less solar radiation, or even raining days. It is not convincing and conclusive to derive something simply based the correlation between SO4 and EC under two categories of relative humidity.

In the Section, the authors talked about wet scavenging. But the authors didn't mention the precipitation during the period. Also Scavenging coefficient for SO4 and EC is also different.

Line 346-347: This is well known that due to the much more solar photons available during summer time, the atmospheric oxidation capacity is much higher than winder season. So much more percentage of SO2 is oxidized. There is really no need to demonstrate this again and again. Line 347-349: I don't see the justification.

In general, I didn't see any new knowledge from this section.

5. Nitrate

Nitrate can be in PM2.5 and PM10. For PM2.5, it is mainly formed from the reaction of nitric acid with NH3. Whether it is an NH3 ample region, it depends on the ratio of NO3/(NO3+HNO3). Unfortunately, in this study, the authors didn't measure HNO3. For PM2.5, nitrate should be dominantly associated with NH4, as Ca(NH4)2 or NaNH4 is

mainly PM10.

Line 390-409: Again, the text in this part has too many "maybe"s. 1. HNO3 is formed through the oxidation of NOx. The authors didn't provide an explanation why relative humidity affects the oxidation of NOx.

Even HNO3 is formed in the air, whether it can be neutralized to nitrate depends on other factors: temperature, availability of NH3.

Line 410-437: Again, the authors used too many "maybe"s to explain the correlation between NO3 and NO2. The authors used "Excess NH4+" to explain the formation of nitrate. Actually, this so called "Excess NH4+" is the NH4 associated with nitrate. The "Excess NH4+" is not the reason, but the results. High concentration of PM2.5 nitrate is definitely associated with high "Excess NH4+", as NH4NO3 is the dominantly PM2.5 nitrate. The dependence of NH4NO3 on temperature is well known. No need to demonstrate it again with another observation. The reasoning in this part is not acceptable. For oxidation of NOx to HNO3, and then formation of nitrate is a complicated process involving gas-phase oxidation by O3 and other photochemical reactions. The authors are suggested to consulate some recent publication, eg. Hauglustaine et al. (2014, ACP), Chen et al. (2016, ACP) and many others.

6. Regional contributions

What is fire map?

Do the authors want to emphasis that the smog events are due to low wind and stable meteorological conditions, and therefore it was mainly due to local emission; or the authors want to say that it was mainly due to transportation? The results presented in this Section were very confusing. In general, the results presented in this paper are very poorly analyzed. There are too many "maybe"s instead of solid and rigorous analysis. I didn't find any valuable new information. A solid and rigorous analysis the formation of a haze event requires a comprehensive air quality model simulation with proper emis-

sion input, comprehensive meteorology, and comparison with observations. Instead the authors tried to use some very basic meteorological variable, mainly relative humidity to explain the formation of three smog events. I don't see any values that merit publication in ACP. I firmly suggest the paper be rejected.

7. Summary and conclusions

The authors are supposed to provide their main findings in this Section. If the authors feel they have produced some new findings, they should emphasize them here.

---

## Referee Comment (RC3) · Anonymous Referee #3 · 12 Mar 2018

Li et al.,

In this manuscript, the author presents two field campaigns in Zibo, Shandong (China) during summer and wintertime (Jan 15-25, 2015 and July 14-31, respectively), and a series measurements were conducted including PM2.5, sulfate, nitrate, OC, EC, as well as gases NOx, SO2, CO, and O3; the authors also provided information on wind speed, wind direction, pressure, temperature and relative humidity. The aim of the authors listed in Introduction was to characterize PM2.5 including chemical composition, diurnal process formation, as well as regional contribution. However, in the title part, the authors promised the characterization of haze pollution, which should have focused

on chemical composition of PM2.5 and the corresponding light properties. The authors also attempted to investigate seasonal variation during summer and wintertime, but two weeks (at most) observations were inadequate. Through the text, the authors tried to illustrate the effect of mixed layer height, photochemical activity, and relative humidity. Unfortunately, the conclusion of each part was not clear enough and somehow already well-known. The way of discussion in Section 3.3, and 3.4 were unacceptable. Hereby, the reviewer would suggest the manuscript be declined for publishing in a journal like ACP. The manuscript should be improved and submitted to a journal for air pollution characterization.

Major comments: 1. According to the content of the text, Title part should be modified to talk about the chemical composition, diurnal profiles, and formation instead of "haze pollution". In addition, the discussion of "source" of the haze was very weak.

2. Abstract is lack of information. The audience would only know the average PM2.5 concentration during the summer and winter observation. No novel information could be obtained in the following part because the behaviors of SIA formation could be explained by known mechanisms.

3. Introduction. The summary of current understanding of air pollution and formation mechanism was insufficient, no scientific question was proposed, no hypothesis was drawn, and why the work should be done was unclear. The authors should at least give the information that why Zibo is important, and what scientific question(s) could be solved through this work.

4. Methodology. Description of quality control/ quality assurance could not be found in this part. The comparison between the online and off-line result would be interesting but they were not provided. Thus the reviewer could not determine the reliability and quality of data presented.

5. Section 3.1. Much information was provided in this part. However, it was no more than a report of local air quality.

6. Section 3.2. The diurnal behaviors of major pollutants are interesting. However, the description and explanation in this part were too general and ambiguous. Indeed, well-known that mixed layer height would affect concentration of pollutants. The reviewer would suggest a study on the diurnal pattern of a ratio of a pollutant over an inactive primary pollutant, e.g. sulfate/EC for a clearer understanding the atmospheric process of pollutants. Moreover, there were numerous studies using aerodyne AMS on NCP, the referee would recommend comparisons between this work and previous studies.

7. 3.3 Sulfate. The referee does not understand the logic of this part. It is NOT surprising AT ALL for a weak correlation between EC and sulfate because one is primary and the other is mainly secondary. The effect of atmospheric dilution due to the shift of boundary layer height could be excluded only when a ratio of pollutant/EC (e.g. sulfate/EC) is adopted. A correlation analysis between sulfate and EC is not meaningful. Line 317-318 is not clear and lack of evidence.

8. 3.4 Nitrate. It is obscure and arbitrary to only use a ratio of $[NH_4^+]/[SO_4^{2-}] = 1.5$ to define an "Excess $NH_4^+$". Moreover, an "Excess $NH_4^+$" is actually from the excess of $NH_3$ which can turn into particulate $NH_4^+$, as a result, NOT a cause. It would go without saying that $HNO_3$ and $NH_3$ reacting in the gas phase and the subsequent portioning is the major source of secondary $NH_4NO_3$ in the particle phase. Figure 7. Why don't you provide a NOR ratio against RH or temperature here? It would be clearer and more straightforward.

9. Section 3.5. Nothing special is drawn in this part.

Please also note the supplement to this comment:
https://www.atmos-chem-phys-discuss.net/acp-2018-83/acp-2018-83-RC3-supplement.pdf